# An Effective Microcurrent Stimulation Method for Inducing Non-Pharmacological Parasympathetic Nervous System Activity for Pain Relief

**DOI:** 10.3390/bioengineering13010052

**Published:** 2025-12-31

**Authors:** Daechang Kim, Jaeeun Ko, Sungmin Kim

**Affiliations:** 1Department of Medical Biotechnology, Bio Medi Campus, Dongguk University, Goyang-si 10326, Republic of Korea; kimdaechang@dongguk.edu; 2Department of Regulatory Science for Medical Devices, Dongguk University, 32, Dongguk ro, Ilsandonggu, Goyang-si 10326, Republic of Korea; je.ko@dgu.ac.kr

**Keywords:** electrocardiogram, parasympathetic nervous system, microcurrent stimulation, pain

## Abstract

This study aims to propose a non-pharmacological approach to pain relief by analyzing changes in electrocardiogram (ECG) parameters following transcutaneous microcurrent stimulation generated according to the pulse train characteristics of intensity and frequency. Therefore, we analyze and interpret stimulation methods that induce parasympathetic nervous system (PNS) activity, which is the clinical basis for pain relief. There were 14 male participants, with a height of 176.08 ± 7.05 cm, a weight of 77.07 ± 10.32 Kg, and an age of 26.35 ± 1.71 years, and 10 female participants, with a height of 160.6 ± 5.88 cm, a weight of 52.9 ± 9.03 Kg, and an age of 24 ± 1.61 years. The microcurrent stimulation patch was attached to the left wrist. In order to observe the PNS induction effect of the measured electrocardiograms, time and frequency domains were analyzed and additional nonlinear analysis was performed. Data measurements had a rest period of more than 1 h depending on the intensity, and more than 1 day depending on the frequency to ensure sufficient stabilization time. Although physiological changes were shown differently in various pulse trains, among them, after 7 Vpp microcurrent stimulation at 1 Hz, the values of the square root of the mean squared differences of successive R-R intervals and instantaneous RR interval variability, which indicate PNS activity in the subjects, significantly increased from 41.31 ± 34.13, 29.23 ± 24.14 ms to 65.09 ± 32.46, 44.56 ± 37.92 ms (*p* < 0.05). Activation of PNS, which can relieve pain, was confirmed only in the 7 Vpp with 1 Hz stimulation. This suggests that microcurrent stimulation can relieve pain in a non-pharmacological way by inducing activation of PNS.

## 1. Introduction

Pain is a multidimensional response generated by neural networks in the brain based on neurological theory [1]. Pain is divided into two types. The first type is psychological protection. Pain, which is activated by detecting noxious stimulation and sense, is an immediate response that protects the nervous system and tissues inside and outside the body from environmental changes. The second is pathological pain caused by abnormal functioning of the nervous system [2]. This type is referred to as nociplastic pain. The International Association for the Study of Pain, a research organization on pain, has identified nociplastic pain as the biggest cause of chronic pain that can be caused by abnormalities in the central nervous system (CNS) and autonomic nervous system (ANS). The biggest problem with nociplastic pain is that it causes maladaptive pain responses in individuals, which significantly reduces quality of life and leads to substantial socioeconomic burden [3,4,5].

One of the dangers of pain is that chronic pain increases the risk of cardiovascular disease [6]. Normally, sensory information such as pain is transmitted to the spinal cord, and the spinal cord transmits various sensory information to the spinal trigeminal nucleus (SN) located in the medulla of the brainstem [7,8]. The transmitted information induces homeostasis of ANS through the nucleus of the solitary tract (NST), which is the integration site of afferent signals in the brainstem, and forms a response for human stability [9,10]. In general, the sympathetic nervous system (SNS), which belongs to the ANS, transmits responses generated through the sympathetic chain, a bundle of nerve fibers located in the anterior part of the spinal cord, to the organs in the body [11,12,13,14]. However, persistent pain caused by pathological damage causes excessive SNS activity, which in turn causes abnormal responses and adverse effects in the cardiovascular system, which is directly connected to the SNS [15,16,17].

Accordingly, research is being conducted on technologies that can provide therapeutic interventions to the nervous system to resolve pain. A common individual therapeutic plan is to use medication. Pharmacological treatment uses analgesics, opiates, and antidepressants. Their role is to block pain signals or activate or deactivate specific nerve systems of the ANS [18,19]. However, cardiovascular and physiological side effects may occur due to excessive activation and inactivity of the nervous system and unwanted interactions caused by drug use [10,11,12,13,14,15,16,17,18,19,20,21,22,23]. Additionally, because drug use may depending on the regulations in different countries new treatment approaches are emerging to address these shortcomings. The effectiveness of these methods can be easily confirmed through heart rate variability (HRV), which can monitor the sympathetic and parasympathetic nervous systems (PNS) that regulate pain conditions, and meta-analyses using HRV have highlighted the need for activation and modulation of the PNS for pain relief [24,25,26].

In this study, we used transcutaneous electrical nerve stimulation (TENS), a novel therapeutic intervention for pain relief. TENS is a medical device that uses microcurrent stimulation with safety approval, and research on pain relief is actively being conducted [27,28,29]. Most TENS stimulation targets the free ectodermal nerve endings of the epidermis (Adelta and C) and produces immediate effects [29]. However, the effectiveness of TENS varies from person to person, with various variables such as the condition, thickness, and intensity of the epidermis, resulting in different physiological responses depending on the location, frequency, and intensity of the stimulation [30]. Therefore, this study aims to analyze ANS response based on the pulse train characteristics generated according to the stimulation frequency and intensity of microcurrent stimulation used in TENS technology. Finally, we propose an effective stimulation method that can induce PNS adjustment and activation, which represents pain relief. It is expected that these studies will be utilized as a safe and effective treatment method to relieve individual pain and prevent abnormal SNS activity and cardiovascular diseases caused by pain.

## 2. Method

### 2.1. Selecting the Location of Stimulatio

This study used electrocardiography (ECG) to analyze ANS response according to the pulse train characteristics of microcurrent stimulation used in TENS. ANS is a component of the peripheral nervous system (PNS) that controls muscle contraction, internal organ activity, and bodily functions, and can directly affect the heart rate and blood pressure of the cardiovascular system. Therefore, the ECG is used as a clinical indicator to evaluate and observe changes in ANS [31,32,33]. The stimulation sites where the pulse train is applied are based on anatomical studies of the spinal nerves and sympathetic chain. The spinal nerves are a bundle of nerves that extend from the spinal cord and spread throughout the body. They, are mixed nerves consisting of a total of 31 pairs. Depending on their location, spinal nerves are classified into cervical nerves, thoracic nerves, and lumbar nerves. These spinal nerves connect to the anterior sympathetic chain through the ventral ramus [34].

It is connected to the sympathetic cardiac chain in the C3–C7 region of the lower cervical nerves and the T1 region of the thoracic nerves, which are the connecting pathways of the ventral ramus, and shows the activation of the sympathetic nervous system by sensory stimulation [35,36]. Additionally, the CT ganglion, consisting of the C3–C7 and T1 nerves, is connected to the pericardium, myocardium, and veins through sympathetic branches, exerting a wide influence on cardiac activity and movement. That is, spinal nerves transmit stimulation information starting from sensory receptors activated by stimulation to the spinal cord and ascending to the CNS. The transmitted information induces activation of the somatosensory cortex, and ANS is regulated by the NST. Finally, changes in the ECG are induced based on the descending exercise information [37,38,39,40]. Therefore, the inner wrist area, where the lower cervical nerves and thoracic nerves can be easily stimulated, was selected as the stimulation location (Figure 1).

### 2.2. Pulse Train Characteristics and Stimulation Procedure

This study was conducted in accordance with DUIRB-202308-05, which was approved by the Dongguk University Institutional Review Board. All participants in the study gave prior consent for the use and publication of data according to the purpose of the experiment. The study design included participants in their 20 s to 50 s without a diagnosis of heart disease or mental illness, and excluded participants taking medications, those with ANS disorders, and those at high risk for stimulation. There were a total of 24 participants, 14 males and 10 females, with demographic statistics shown in Table 1.

The pulse train characteristics of the microcurrent stimulation used to analyze ANS response of the participant used static stimulation of 1 Hz and 40 Hz included in the low and high frequency ranges (Figure 2a,b). Additionally, we performed additional analyses of 5–40 Hz dynamic stimulation sequentially increasing from low to high frequency ranges (Figure 2c). Each stimulation had an intensity of 3, 7, and 11 peak to peak voltage. Data for the stimulation were measured sequentially according to intensity magnitude. That is, nine data measurements were taken per subject. The TENS equipment used was LT-1803 (Shenzhen Geniuschip Electronic Co., Ltd., China) with various pulse train patterns. ECGs were collected using the BioPac System MP36, which is used for clinical index analysis in physiology studies and cardiovascular disease analysis. Measurements were performed using SS57L EDA leads and BIOPAC SYSTEMS EL500 2 Electrodes patches (Manufacturer: BIOPAC Systems, Inc., Goleta, CA, USA). The ECG measurement method used the Lead II method to minimize noise that may be caused by microcurrent stimulation. The data measurement location and method are as shown in Figure 3.

All participants had a stimulation patch attached to their left wrist. For ECG measurement, the leads were attached with a positive electrode on the left leg, a ground electrode on the left leg, and a negative electrode on the right wrist. Before the measurement, the participants sat on a chair and rested for about 3 min to stabilize the cardiovascular and ANS changes that may occur due to movement. First, the ECG was measured in a relaxed state for 2 min. Afterwards, microcurrent stimulation was applied for 15 min. Additional ECGs were measured at 0–2, 5–7, and 10–12 min during stimulation to obtain data for the early, middle, and late stimulation periods. Finally, after the stimulation was terminated, the ECG was measured for an additional 2 min to observe the responses and changes due to the stimulation. The measurement interval according to intensity in a pulse train with the same frequency characteristics is approximately 1 to 2 h. However, the measurement interval according to frequency characteristics was set to approximately 3 to 4 days to provide a sufficient recovery phase.

### 2.3. Data Parameter Analysis Method

HRV, a method of analyzing the difference in intervals between heart beats in a measured electrocardiogram, is used as a clinical indicator of the regulation of ANS. HRV can analyze changes in the sympathetic and parasympathetic nervous systems in the time and frequency domains [41,42]. The parameters of HRV used are as shown in Table 2. The RR interval is, the time elapsed between two successive R-waves of the QRS signal on the electrocardiogram. It represents the variability between heart rates, which is a physiological change that varies depending on the internal and external environment of the human body. Among the parameters calculated through this, the indicators representing SNS are the standard deviation of the normal to normal interval (SDNN), SD2, and 0.05–0.15 Hz normalized low frequency band (LF norm). However, since SDNN can represent the comprehensive activation of the sympathetic and parasympathetic nervous systems, this study analyzed it as the overall activity of ANS [43]. The indices representing PNS are the root mean square of successive differences (RMSSD), SD1, and 0.150.4 Hz normalized high frequency band (HF norm). SD1 means the standard deviation of the Poincaré plot perpendicular to the line of identity, while SD2 represents the standard deviation of the Poincaré plot along the line of identity [44].

Additional analytical methods involve assessing complexity and randomness. Entropy is an indicator of the randomness of time series data and reflects the health of the cardiovascular system and the ANS [45,46]. The analysis of complexity and variability can reveal changes in neurons due to excitatory and inhibitory commands that occur in the CNS, such as stress, alcohol, and functional system decline [47]. Among the important indicators, the detrended fluctuation analysis (DFA) indicator is a method to evaluate the regulatory influence of the cardiovascular system on the function of the ANS in the measured data [48,49,50]. HRV has been used as a reliable indicator of ANS reactivity to pain stimuli, and significant changes in HRV parameters were observed in a study that included a total of 6364 participant samples [51,52,53]. Most importantly, it was confirmed that the higher the PNS activity, the higher the self-regulation and pain suppression ability [53].

Accordingly, the changes in the ANS before and after stimulation and according to stimulation time were divided into a total of five areas and analyzed. All data were statistically examined using a paired t-test to verify whether there were differences between the data before stimulation and the data after microcurrent stimulation. Ultimately, we propose a non-pharmacological method to activate the PNS through the response of the autonomic nervous system to microcurrent stimulation.

### 2.4. Data Statistical Analysis

Changes in values for all parameters were expressed as mean and standard deviation according to stimulation and were rounded to the third decimal place using NumPy’s numpy.round function. Finally, differences in parameters by stimulus were verified using one-way repeated measures ANOVA and paired *t*-test simultaneously. Therefore, it is possible to meaningfully observe the average difference according to the stimulus in the same subject and the change in the value of data measured repeatedly over time.

## 3. Results

### 3.1. Nervous System Response to Microcurrent Stimulation

Changes in DFA to HRV parameters due to microcurrent stimulation for all pulse train characteristics are shown in Table 3. A common feature is that a significant decrease in DFA occurs in situations where microcurrent stimulation is delivered (*p <* 0.05). However, if DFA changes occur within the range of 1.0–1.5, it means that flexible response and adaptation to stimulation occurs in the ANS [54,55]. Common characteristics of stimulation are the method delivery and pathway. Microcurrent stimulation transmits sensory information to the spinal nerves through sensory receptors and ascends to the brainstem. After that, the movement of the organs in the body is coordinated through the descending motor information based on the processed information. That is, DFA shows that the ascending and descending information transmission is in operation and can indicate cardiovascular changes due to the adjustment of the ANS. These results suggest that DFA can serve as a criterion for determining whether microcurrent stimulation is being performed normally.

### 3.2. Stimulation Methods for Parasympathetic Nervous System Activity

As shown in Table 4, 7 Vpp microcurrent stimulation with 1 Hz pulse train characteristics showed significant increases in RMSSD. SD1 indicates the PNS after stimulation. As the stimulation progresses, this suggests that the ANS is activated by sufficient stimulus intensity and adaptation by NST is induced [56,57,58,59]. In other words, this shows that PNS activation to enhance self-regulation and pain suppression ability can be induced by microcurrent stimulation with 1 Hz pulse train characteristics at 7 Vpp stimulation intensity.

### 3.3. Parameter Changes According to Frequency Characteristics

We analyzed the frequency dependent parameter changes at 7 Vpp microcurrent stimulation intensity, which has the PNS activation effect. Microcurrent stimulation at 1 and 40 Hz with static characteristics produced similar patterns of increase, but 1 Hz microcurrent stimulation better demonstrated progressive ANS activation. We confirmed that this gradual activation of the ANS could induce significant increases in RMSSD and SD1, which represent the PNS, after a 1 Hz type of stimulation. However, the 40 Hz pulse train belonging to the high-frequency range shows a pattern similar to that of 1 Hz, but shows a comprehensive ANS activation pattern rather than PNS activation.

Next, pulse train characteristics that dynamically vary from 5 Hz to 40 Hz indicate rapid activation of the ANS upon initial stimulation and increased PNS parameters. Ultimately, however, no significant changes in the parameters representing the PNS were found. Additionally, it was confirmed that LF decreased when microcurrent stimulation with dynamically changing 5–40 Hz pulse train characteristics was applied. Previous studies have confirmed that pain caused by persistent stimulation decreases LF activity and increases nervous system fatigue [60,61,62,63]. These results indicate that static stimulation in the low frequency region can more effectively induce ANS responses and adaptations for PNS activation than variable and high frequency stimulation.

## 4. Discussion

Among the occurrences of pain, pathological pain caused by abnormal function of the nervous system is a disease that causes chronic pain and significantly reduces an individual’s quality of life. To solve this, the aim of this study was to analyze the physiological responses of microcurrent stimulation using TENS to propose pulse train characteristics that can be used as non-pharmacological nerve stimulation to treat pain. Therefore, we analyzed the autonomic nervous system response according to the pulse train type and intensity using electrocardiography and proposed the characteristics and intensity of the pulse train that can activate the parasympathetic nervous system for pain relief [53,64,65].

First, a common feature across all stimulation was that DFA was significantly reduced when the wrist was stimulated using microcurrent stimulation. In this study, we focus on the fact that among the methods of stimulation delivery and the path of stimulation, an increase in blood pressure caused by muscle contraction due to external stimulation can induce cardiovascular changes [66,67,68]. Normal muscle contraction is a response to muscle sympathetic nerve activity that occurs due to the activation of SNS, which promotes circulatory function and causes muscle contraction [69,70,71]. The resulting increase in blood pressure may reduce HRV [72,73,74]. That is, microcurrent stimulation induces changes in blood pressure using muscle contraction, and transmits the senses detected through baroreceptors and mechanoreceptors, which are sensory receptors present in blood vessels, to the brainstem so that information can be processed. Therefore, this suggests that changes in the autonomic nervous system are induced and HRV parameters can change, and it can be confirmed that it returns to the original state when the stimulation ends.

The primary parameter influencing neural stimulation among the pulse_train characteristics is the stimulation intensity. The medulla oblongata connected to the spinal nerves has a total of four nerve nuclei, among which the SN is the nerve nucleus that primarily transmits information such as touch and pain [12]. The SN contains a nociceptive innervation that processes mechanical and physical stimulation intensity and noxious stimulation in response to external stimuli, and plays a role in coordinating the response of the autonomic nervous system together with the NST. However, as shown in the results, it is judged that low intensity stimulation of 3 Vpp does not induce a response from the ANS [8,75,76]. Conversely, a high intensity, strong stimulation of 11 Vpp did not produce any significant results through protective effects. SN forms a descending pain modulation pathway through the reticular activating system formed in the brainstem and plays an important role in inhibiting pain signals from the spinal cord [77,78]. However, this inhibitory response decreases the ability of the autonomic nervous system to regulate the cardiovascular system, as evidenced by a significant decrease in the LF norm at 5–40 Hz stimulation in Table 5. That is, a significant decrease in LF norm can be confirmed as a change expressed as pain.

The most interesting finding from this study is the microcurrent stimulation with a 1 Hz pulse train of 7 Vpp. This stimulation showed a significant increase in SDNN and SD2, which represent the overall activity of the ANS, in the 0–2 and 4–6 min stimulations, and showed parasympathetic nervous system dominance activity after the stimulation. The PNS suppresses inflammation, stress, SNS activation, and has analgesic effects on the production of opioid neurotransmitters. That is, this suggests that PNS activity may be a target for new pain treatments [65,79,80].

A variety of studies are investigating ANS reactivity using non-pharmacological methods that can replace pharmacological methods. The 1 Hz pulse train characteristic shown in the results of this study is also used in repetitive transcranial magnetic stimulation, which also shows the influence of PNS dominance on the participants of non-pharmacological methods. One of these features is that, depending on the stimulus, the cerebral cortex of the central nervous system can selectively collect neural responses, expanding the response size of physiologically matched neurons [81,82]. That is, it is judged that the greatest responsiveness was shown at the 1 Hz stimulation, which is most similar to the heartbeat and breathing patterns, which are the average cardiovascular movements that occur in the coordination of the ANS [83,84].

This study has three limitations. The first is that the analysis area for the stimulation response is short. In general studies, the HRV analysis area for cardiovascular analysis is about 5 min. However, various studies are conducting analyses on ultra-short-term areas of at least 1 min, but entropy and DFA, which observe cardiovascular variability, are set to 2 min, so this study was set to 2 min. Second, not all of the subjects’ ANS were controlled equally. The responses and activity of the autonomic nervous system can vary according to circadian rhythms, as well as depending on food intake, environmental conditions, and psychological factors. However, to minimize these variations, we established basic control measures, such as checking the individual’s status, drinking, and smoking prior to measurement. In addition, measurements were not taken at 12–16 o’clock, when the sympathetic nerve activity of the ANS can be the greatest. Finally, additional research on the epidermis is needed by observing and analyzing the final autonomic nervous system changes without evaluating the effects of epidermal cells that may occur in the epidermis of the wrist, which is the stimulation site of TENS.

## 5. Conclusions

This study analyzed ANS response according to the pulse train characteristics of microcurrent stimulation used in the easily usable TENS technique, and confirmed that PNS activity can be induced as a non-pharmacological stimulation method. Finally, we confirm that stimulating the wrist with a 1 Hz pulse train characteristic at a 7 Vpp stimulation intensity can induce PNS activation. A major advantage of TENS is its compatibility with pharmacological treatments and other integrated therapeutic approaches, so it is expected to provide greater effects in pain relief. 

## Figures and Tables

**Figure 1 bioengineering-13-00052-f001:**
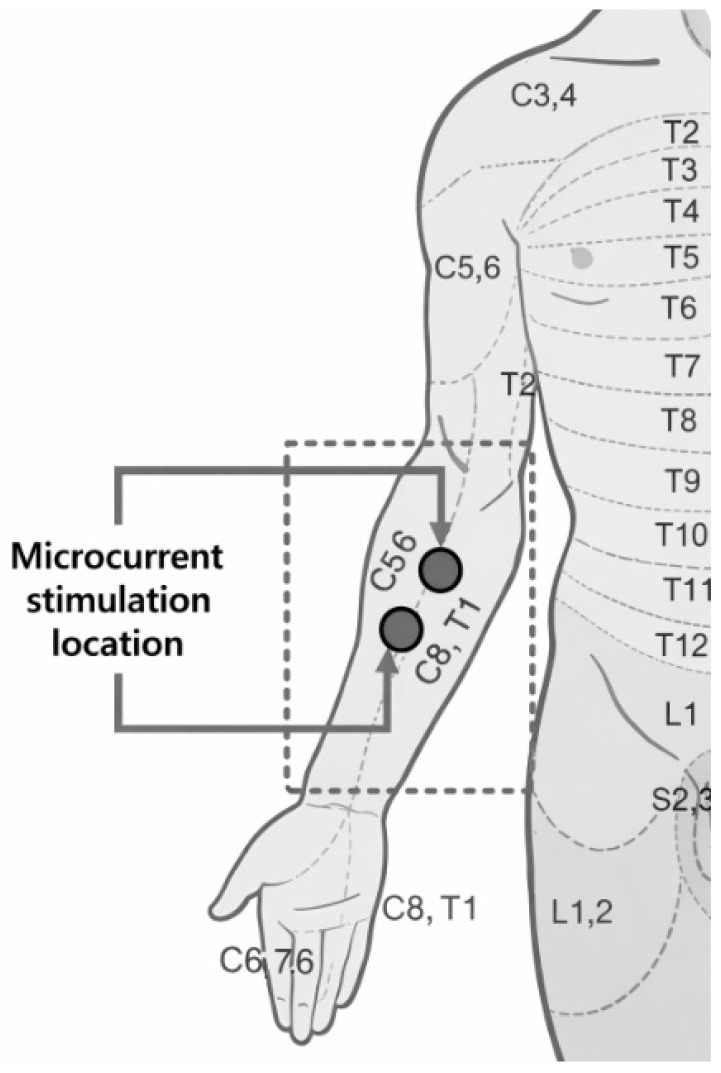
Microcurrent stimulation selection location and connected spinal nerves.

**Figure 2 bioengineering-13-00052-f002:**
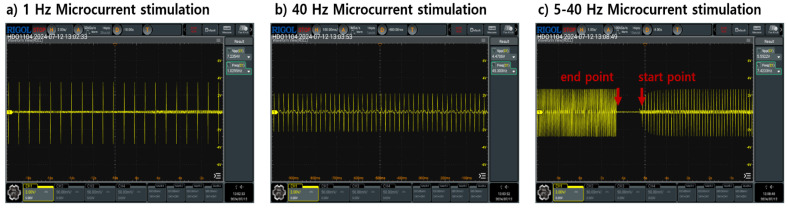
Type of pulse train used for stimulation (**a**) 1 Hz pulse train type, (**b**) 40 Hz pulse train type, (**c**) 5–40 Hz pulse train type.

**Figure 3 bioengineering-13-00052-f003:**
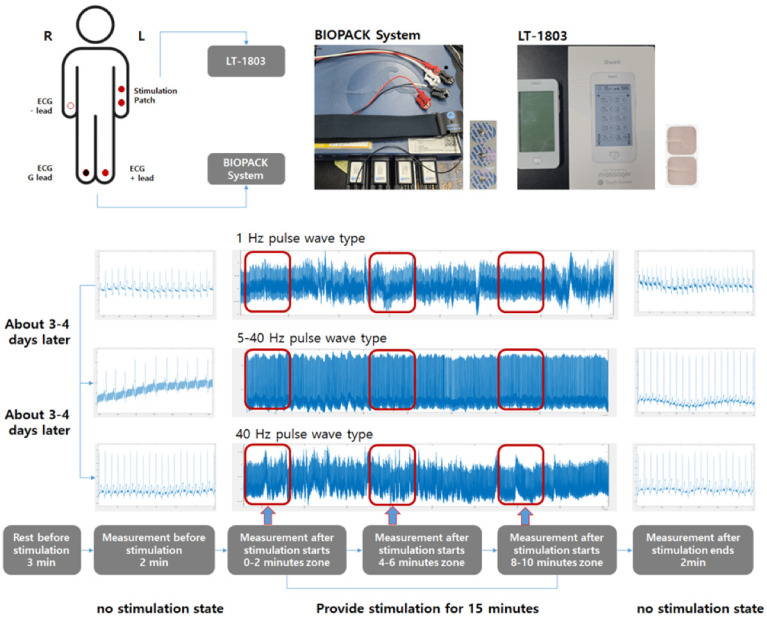
Patch attachment location and data measurement procedure.

**Table 1 bioengineering-13-00052-t001:** Demographic characteristics of participants.

Participants (Count)	24			
	Male	14	Female	10
Height (Cm mean ± std)	176.08 ± 7.05		160.6 ± 5.88	
Weight (Kg mean ± std)	77.07 ± 10.32		52.9 ± 9.03	
Age (Years mean ± std)	26.35 ± 1.71		24 ± 1.61	

**Table 2 bioengineering-13-00052-t002:** Autonomic nervous system observation parameters using heart rate variability.

Parameter	Formula	Analyze
RR-interval	${peak}_{n-1}-{peak}_{n}$	Heart rate variability
The standard deviation of normal to normal R-R intervals (SDNN)	$\sqrt{\frac{1}{n}\sum_{i=1}^{n}{\left({RR}_{i}-\bar{RR}\right)}^{2}}$	Sympathetic/parasympathetic nerves
The square root of the mean squared differences of successive R-R intervals (RMSSD)	$\sqrt{\frac{1}{n-1}\sum_{i=1}^{n-1}{\left({RR}_{i+1}-{RR}_{i}\right)}^{2}}$	Parasympathetic nerves
Instantaneous RR interval variability (SD1)	$\frac{RMSSD}{\sqrt{2}}$	Short term volatility
Continuous RR interval variability (SD2)	$\sqrt{2{SDNN}^{2}-\frac{1}{2}}{RMSSD}^{2}$	Long term volatility
Entropy	$-log(\frac{{RR}_{i+1:n}}{{RR}_{i:n-1}})$	Randomness
Detrended fluctuation analysis (DFA)	$Y\left(i\right)=\sum_{i=1}^{n}\left({RR}_{i}-\bar{RR}\right)$ $F\left(n\right)=\sqrt{\frac{1}{n}\sum_{i=1}^{n}{\left(Y\left(i\right)-Y_{fit,n}\left(i\right)\right)}^{2}}$ $F\left(n\right)\sim n^{\alpha }$ α is DFA	Long-term correlation
Low pass band power spectral density (LF norm)	${PSD}_{0.05-0.15Hz}/{PSD}_{0.05-0.45Hz}$	Sympathetic/parasympathetic nerves
High pass band power spectral density (HF norm)	${PSD}_{0.15-0.4Hz}/{PSD}_{0.05-0.45Hz}$	Parasympathetic nerves
LF/HF	${LF}_{norm}/{HF}_{norm}$	Autonomic nervous system balance

**Table 3 bioengineering-13-00052-t003:** Changes in determined fluctuation analysis by microcurrent stimulation. * is *p* < 0.05.

Hz	Vpp	Before Stimulus	Stimulus 0–2 min Interval	Stimulus 4–6 min Interval	Stimulus 8–10 min Interval	End of Stimulus
1	3	1.23 ± 0.24	1.12 ± 0.19 *	1.11 ± 0.17 *	1.12 ± 0.19 *	1.23 ± 0.29
	7	1.37 ± 0.23	1.14 ± 0.16 *	1.14 ± 0.17 *	1.16 ± 0.14 *	1.31 ± 0.33
	11	1.32 ± 0.28	1.11 ± 0.17 *	1.15 ± 0.16 *	1.14 ± 0.19 *	1.25 ± 0.20
40	3	1.26 ± 0.19	1.07 ± 0.16 *	1.09 ± 0.16 *	1.09 ± 0.15 *	1.20 ± 0.20
	7	1.22 ± 0.20	1.09 ± 0.16 *	1.10 ± 0.17 *	1.09 ± 0.17 *	1.24 ± 0.20
	11	1.28 ± 0.25	1.06 ± 0.16 *	1.10 ± 0.14 *	1.10 ± 0.16 *	1.21 ± 0.22
5–40	3	1.19 ± 0.20	1.06 ± 0.13 *	1.06 ± 0.16 *	1.07 ± 0.18 *	1.16 ± 0.22
	7	1.22 ± 0.27	1.04 ± 0.16 *	1.03 ± 0.16 *	1.09 ± 0.18 *	1.15 ± 0.21
	11	1.25 ± 0.23	1.05 ± 0.15 *	1.08 ± 0.16 *	1.10 ± 0.18 *	1.15 ± 0.19

**Table 4 bioengineering-13-00052-t004:** Changes in parasympathetic nerves before and after microcurrent stimulation * is *p* < 0.05.

	3 Vpp		7 Vpp		11 Vpp	
	RMSSD					
Hz	Before Stimulus	End of Stimulus	Before Stimulus	End of Stimulus	Before Stimulus	End of Stimulus
1	74.18 ± 63.85	75.36 ± 49.70	41.31 ± 34.13	65.09 ± 32.46 *	47.04 ± 50.32	58.45 ± 23.78
40	47.00 ± 38.28	41.00 ± 33.41	38.25 ± 17.98	41.50 ± 41.05	37.33 ± 22.71	45.25 ± 41.32
5–40	47.79 ± 37.13	36.95 ± 24.77	44.45 ± 31.05	43.66 ± 23.50	47.91 ± 31.24	47.70 ± 31.06
	SD1					
1	52.49 ± 45.08	53.28 ± 35.13	29.23 ± 24.14	44.56 ± 37.92 *	33.15 ± 35.54	38.05 ± 27.86
40	33.22 ± 27.04	28.96 ± 23.56	27.06 ± 12.77	29.34 ± 28.98	26.41 ± 16.00	32.04 ± 29.27
5–40	33.83 ± 26.28	26.11 ± 17.45	31.40 ± 21.95	30.79 ± 16.58	33.85 ± 22.04	33.72 ± 21.95

**Table 5 bioengineering-13-00052-t005:** Changes in heart rate variability parameters by 7 Vpp microcurrent stimulation according to frequency characteristics. * is *p* < 0.05.

Type	Parameters	Before Stimulus	Stimulus 0–2 min Interval	Stimulus 4–6 min Interval	Stimulus 8–10 min Interval	End of Stimulus
1 Hz	RR-intervals (s)	0.75 ± 0.08	0.74 ± 0.07	0.74 ± 0.08	0.74 ± 0.08	0.75 ± 0.07
	SDNN (ms)	54.68 ± 22.44	61.04 ± 19.16 *	63.5 ± 28.35	60.40 ± 23.99	65.09 ± 32.46
	RMSSD (ms)	41.31 ± 34.13	47.68 ± 33.08	50.45 ± 43.38	60.40 ± 23.99	65.09 ± 32.46 *
	SD1 (ms)	29.23 ± 24.14	33.62 ± 23.43	35.69 ± 30.62	34.02 ± 27.46	44.56 ± 37.92 *
	SD2 (ms)	70.40 ± 24.06	77.77 ± 20.73	80.72 ± 31.00 *	76.90 ± 24.76	78.07 ± 32.95
	Entropy	1.27 ± 0.34	1.21 ± 0.35	1.15 ± 0.31 *	0.20 ± 0.31	1.26 ± 0.44
	DFA	1.37 ± 0.23	1.14 ± 0.16 *	1.14 ± 0.17 *	1.16 ± 0.14 *	1.31 ± 0.33
	LF norm	0.38 ± 0.13	0.36 ± 0.15	0.36 ± 0.18	0.37 ± 0.13	0.36 ± 0.14
	HF norm	0.65 ± 0.56	0.73 ± 0.74	0.95 ± 1.15	0.63 ± 0.58	1.06 ± 1.02
	LF/HF	3.48 ± 3.77	5.50 ± 7.84	5.99 ± 8.11	4.36 ± 5.22	2.99 ± 3.30
40 Hz	RR-intervals (s)	0.76 ± 0.10	0.76 ± 0.10	0.76 ± 0.09	0.76 ± 0.10	0.75 ± 0.08
	SDNN (ms)	49.41 ± 0.15	53.62 ± 14.31	53.54 ± 15.06	52.62 ± 16.60	57.87 ± 26.91
	RMSSD (ms)	38.25 ± 17.98	37.37 ± 13.88	37.93 ± 18.79	36.41 ± 17.68	41.50 ± 41.05
	SD1 (ms)	27.06 ± 12.77	26.41 ± 9.81	26.82 ± 13.28	25.81 ± 12.04	29.34 ± 28.98
	SD2 (ms)	64.10 ± 19.84	70.78 ± 18.98	70.04 ± 19.41	59.47 ± 21.19	75.01 ± 28.35
	Entropy	1.50 ± 0.43	1.36 ± 0.28	1.35 ± 0.28	1.39 ± 0.31	1.38 ± 0.33
	DFA	1.22 ± 0.20	1.09 ± 0.16 *	1.10 ± 0.17 *	1.09 ± 0.17 *	1.24 ± 0.20
	LF norm	0.40 ± 0.14	0.36 ± 0.16	0.40 ± 0.17	0.39 ± 0.17	0.37 ± 0.17
	HF norm	0.69 ± 0.53	0.68 ± 0.52	0.70 ± 0.76	0.68 ± 0.69	0.62 ± 0.66
	LF/HF	2.84 ± 2.68	3.76 ± 4.54	4.58 ± 4.71	4.29 ± 4.53	4.24 ± 4.19
5–40 Hz	RR-intervals (s)	0.79 ± 0.11	0.80 ± 0.10	0.79 ± 0.10	0.79 ± 0.09	0.80 ± 0.10
	SDNN (ms)	53.62 ± 19.02	73.12 ± 33.58 *	61.50 ± 27.63	56.33 ± 19.46	58.50 ± 20.31
	RMSSD (ms)	44.45 ± 31.05	71.17 ± 55.93 *	60.87 ± 48.27	46.87 ± 31.95	43.66 ± 23.50
	SD1 (ms)	31.40 ± 21.95	50.72 ± 39.58 *	42.99 ± 34.03	33.15 ± 22.57	30.79 ± 16.58
	SD2 (ms)	67.26 ± 21.84	87.02 ± 34.68 *	73.58 ± 25.99	70.89 ± 21.59	75.25 ± 28.04
	Entropy	1.37 ± 0.33	1.21 ± 0.34 *	1.32 ± 0.36	1.36 ± 0.28	1.42 ± 0.29
	DFA	1.22 ± 0.27	1.04 ± 0.16 *	1.03 ± 0.16 *	1.09 ± 0.18 *	1.15 ± 0.21
	LF norm	0.37 ± 0.17	0.22 ± 0.07	0.28 ± 0.09	0.28 ± 0.08	0.30 ± 0.12
	HF norm	1.13 ± 1.28	1.56 ± 1.39	1.22 ± 0.99	1.02 ± 0.91	0.99 ± 0.87
	LF/HF	2.85 ± 2.46	1.30 ± 1.19 *	1.61 ± 1.31 *	2.02 ± 1.59	2.39 ± 2.65

## Data Availability

The data were obtained in conjunction with a government project. For inquiries, please contact the corresponding author. Please clearly state your affiliation, name, purpose of use, etc. All materials can be delivered after approval of the request. This data was not included in the IRB design phase for uploading. Therefore, it is based on USB storage. Therefore, upon request, we propose that data be stored and then delivered upon approval by the principal investigator.

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
