# Peer review of "An Effective Microcurrent Stimulation Method for Inducing Non-Pharmacological Parasympathetic Nervous System Activity for Pain Relief"

_bioengineering, 2025, doi:10.3390/bioengineering13010052_

Round 1
Reviewer 1 Report
Comments and Suggestions for Authors
Proposal of a microcurrent stimulation method to induce non-pharmacological parasympathetic nervous system activity for pain relief through analysis of autonomic nervous system response
Thank you for trusting me to review this paper.
Below, I list the line of the manuscript from which I am making my comment.
Thank you.
Line 46. Include a bibliographic reference for this statement: One of the dangers of pain is that chronic pain increases the risk of cardiovascular disease.
Line 47. Explain and reference the following clarification: Sensory pain occurs via two pathways. On the one hand, the somatosensory pathway (somatic system-peripheral nervous system), and on the other hand, the autonomic sensory pathway through epidermal cells innervated by free nerve endings innervated by the autonomic nervous system, which are ectodermal.
Line 71. There are publications on other transcutaneous neuromodulation techniques. I sent them the link to one of these articles: Selva-Sarzo F, Sánchez Romero EA, Cuenca-Zaldívar JN, García-Haba B, Akiyama C, Sillevis R, Fernández-Carnero S. Effects on perceived pain and somatosensory function after transcutaneous neuromodulation in patients with chronic low back pain: a quasi-experimental study with a crossover intervention. Front Pain Res (Lausanne). 2025 Apr 15;6:1525964. doi: 10.3389/fpain.2025.1525964. PMID: 40303317; PMCID: PMC12037630.
Line 82: The methods are based on peripheral mesodermal innervations, without mentioning that, first and foremost, the free ectodermal nerve endings of the epidermis (Adelta and C) are also stimulated. The epidermis is ectodermal, and the fascia and muscles are mesodermal. Most epidermal cells are stimulated solely through electromagnetic fields such as TENS currents, and their effect is immediate. They also stimulate the mesodermal Abeta fibers. It is the ectodermal epidermal fibers that modulate the autonomic nervous system, not the peripheral mesodermal fibers. The reference I sent you and the articles it cites provide further information on all of this.
Therefore, they should include these two afferent pathways and explain why they did not assess the effects of epidermal cells, including this in the Limitations section, and also acknowledge this bias in the conclusions.
Line 237: Clarify, modify, and explain that the autonomic nervous system is innervated by epidermal free nerve endings and not by mesodermal or peripheral Aβ innervations. I encourage the authors to explain that if changes are observed when measuring changes in peripheral structures, it is necessarily because epidermal nerve endings have transmitted the information directly to the brain and peripheral nerve endings. It would be advisable to include in the Limitations and Conclusions section that epidermal changes should be studied in future research.
Line 246: It is advisable to include the references that also reach the same conclusion regarding transcutaneous neuromodulation.
Author Response
First of all, I would like to thank you for reading the paper and providing a good review.
Comments 1.
Line 46. Include a bibliographic reference for this statement: One of the dangers of pain is that chronic pain increases the risk of cardiovascular disease.
= Added meta-analysis papers with significant associations.
Comments 2.
= First, I sincerely thank you for your excellent explanation. I agree with the reviewer's comments. From the beginning of our experimental design, we considered two potential pathways.
The first is via the spinal cord, and the second is via the cranial nerves.
Further refinement reveals that the brainstem, which includes the somatosensory pathway and the autonomic nervous system, is involved.
However, this paper focuses on the nucleus accumbens in the brainstem, which is known to receive information from both pathways and plays a crucial role in regulating the autonomic nervous system.
Also, I added references.
Comments 3.
= I thoroughly read the paper. It was a time to discover interesting facts and methods through its content. I've added the relevant information as an attachment.
Comments 4.
= We were able to explain our research more effectively by following the explanation. We revised the wording to make it easier to understand by including relevant citations and additional explanations.
Comments 5.
= I've added relevant information and limitations. However, this paper focuses on the nucleus solitarius, an autonomic nervous system center responsible for two types of pain. Since no actual skin-related studies were conducted, I believe adding the skin-related information to the conclusion could be confusing to the authors.
However, I would like to clarify that the information contained in the Introduction and Limitations sections has been added.
= The final limitation was added to emphasize the need for epidermal studies.
Comments 6.
= I added the attached document.

Reviewer 2 Report
Comments and Suggestions for Authors
This paper focuses on the research of inducing parasympathetic nerve activity through transcutaneous microcurrent stimulation to achieve non-pharmacological analgesia, with a clear clinical application orientation. There are some comments:
- The study did not set up a blank control group (without any stimulation) or a sham stimulation control group (only electrodes were attached but no actual microcurrent was output). Due to the subjectivity of pain perception and the possible influence of the placebo effect, the absence of a sham stimulation control group makes it impossible to rule out the interference of psychological factors on the results.
- Only through HRV was it inferred that enhanced parasympathetic nerve activity was related to pain relief, but the pain threshold and pain score were not directly evaluated. Please give some explanation.
- Only the left wrist was selected as the stimulation site, without comparing other potential more effective stimulation locations (such as the ears, neck, etc., which are commonly used for vagus nerve stimulation).
Author Response
First of all, I would like to thank you for reading the paper and providing a good review.
Thank you for asking so many crucial questions. It gave me the opportunity to review the paper once again. Thank you.
Comments 1.
The first factor is parasympathetic nerve activity. The parasympathetic nerve can be activated in the absence of any stimulation, perceived as rest, and this could potentially influence the results and interpretation of the paper.
To address this issue, although not described in the paper, we note that a control group using stimulation other than current stimulation exists.
However, we have outlined the limitations of this study, noting that psychological factors cannot be ruled out.
Comments 2.
This is consistent with our research hypothesis and setup.
The fundamental goal was to determine "what stimulus and intensity can effectively induce parasympathetic nerve stimulation?"
Accordingly, this study used TENS to minimize pain, and therefore pain scores were not directly assessed.
In other words, this study discusses the methodology for achieving the most effective parasympathetic nerve stimulation in TENS, the most commonly used method for pain relief.
However, we added additional literature on potential issues related to parasympathetic nerve activity and pain relief to provide supporting evidence.
Comments 3.
= This is a crucial question.
Currently, there is a method for activating the vagus nerve by microcurrent stimulation of the ear, known as cranial nerve stimulation. This is the area of research we are conducting together.
However, vagus nerve stimulation is currently being studied more deeply in the treatment of diseases other than pain, so it was excluded from this study.
Basic TENS, a method involving superficial stimulation of the skin, has extensive clinical evidence supporting various pain studies.
Furthermore, it is the most commonly used method in the healthcare field, targeting somatic nerves (such as the forearm, thigh, and joints). For these reasons, we selected the left wrist for our initial research.
